# The Perplexing Mental Health Comorbidity of Alice in Wonderland Syndrome (AIWS): A Case Study

Jennings Hernandez 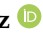

Clinical Sciences Department, Washington University of Health and Science, Columbus, OH 43213, USA; jenningshernandez7@gmail.com

**Abstract:** The Alice in Wonderland Syndrome (AIWS) is an unusual and uncommon condition that falls under the umbrella of neurology and psychiatry. It is characterized by the presence of complex perceptual and visual discord. Additionally, there are visual hallucinations that are multi-dimensional. This syndrome was first described by John Todd in the 1950s, and it was loosely based on the book Alice in Wonderland. A man in his 30s arrived at his doctor's appointment with a chief complaint of a pounding cluster headache that lasted over a full day. In addition, he mentioned that there was an aura preceding his headaches. The pain was so intense, it was debilitating him from routine activities. Before the headaches, he explained that he would sense bizarre physical and visual behaviors. During these episodes, he explained that things around him appeared distorted and of various sizes. Things in his room appeared to be more distant than they really were and larger in size (macropsia and micropsia). He described the fingers on his right hand to be much smaller compared to his left hand (micropsia). Objects around him were deformed and distorted (metamorphopsia). His symptoms lasted 45 minutes. He did not suffer from any previous headaches or hallucinations. He was a healthy man with a clean bill of health as per his medical records. Upon examination, the attending physician described the patient as alert, oriented to time and place, and under no obvious distress. All labs performed returned normal including a 10-panel drug test. These were tested to see if he was under the influence of any narcotic, stimulant, or other substances. The physician prescribed 500 mg of valproic acid to take daily. Three months later during his follow-up, he mentioned his symptoms had subsided but were still present. His dose was again increased to 1000 mg/day, eventually stopping all further symptoms from surfacing. He has not had another episode in three months. The Alice in Wonderland Syndrome is known to be associated with headaches with preceding auras. It is common in the pediatric and adult populations. In this paper, I introduce a case of a patient who displays migraines with preceding auras, indicative of AIWS.

**Keywords:** Alice in Wonderland Syndrome; metamorphopsias; Todd's syndrome; lilliputian hallucinations; macropsia; micropsia; teleopsia; pelopsia; aschematia; dysmetropsia; migraines; aura

## 1. Introduction

The Alice in Wonderland Syndrome can be quite unique and out of the ordinary. It presents a perceptual phenomenon that leads to transitory occurrences of distorted perception and disorientation (metamorphopsias). People who experience this rare condition may undergo brief sensations of feeling larger or smaller than they really are. They can also perceive the room to be much larger or smaller than it really is. The furniture around these individuals may be presumed to be distorted. This condition is a syndrome that affects all the senses and perception. Vision, touch, and hearing are all equally affected. These senses are not the result of issues with the eyes, ears, or hallucinations. Instead, it is alterations in how the brain perceives the environment. In the 1950s, this syndrome was labeled "Alice in Wonderland Syndrome" (AIWS) due to the unusual and extraordinary observations made by patients. Patients with this syndrome described their physical frame as being altered in illusions with changes in forms, dimensions, and motions of objects

(Figures 1 and 2). He suggested these paramnesias and hallucinations coincide with the body alterations the character Alice displayed in Lewis Carroll's earliest Alice's Adventures in Wonderland [1]. These distortions experienced by these patients include metamorphopsia, macropsia, micropsia, teleopsia, and pelopsia. These incidents are perceived as a group of cognitive impairments referred to as aschematia and dysmetropsia. Mastria et al. (2016) stated that this syndrome has many different etiologies; however, EBV infection is the most common cause in children, while migraine effects are more common in adults. Much data support a strict relationship between migraine and AIWS, particularly in children [2]. To date, no ICD-10 or DSM-5 criteria have been confirmed and identified for AIWS. Diagnosis is made on presenting signs and symptoms and ruling out any other diagnosis such as primary psychiatric disorders, viral infections, and central nervous lesions. While it has historically been assumed that this syndrome is rare, epidemiologic studies in patients with migraine have reported an AIWS prevalence rate of up to 15% among this population. Previously published etiologies implicated in this syndrome include viral infections (particularly Epstein-Barr virus [EBV]), migraine, epilepsy, central nervous system lesions, and hallucinogenic substances. One pediatric study found a family history of migraine in nearly 50% of patients. This suggests a possible genetic predisposition to this syndrome [3].

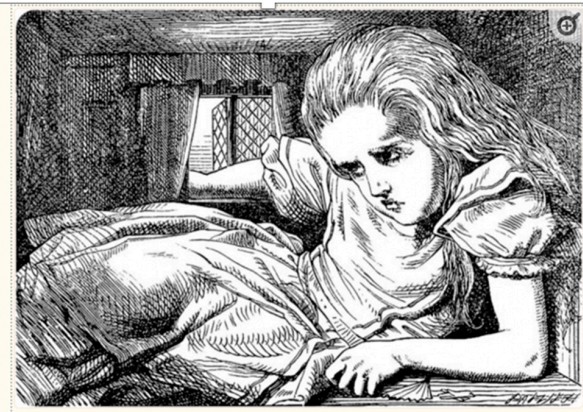

**Figure 1.** Alice experiences total-body macrosomatognosia. Illustrated by John Tenniel (1865).

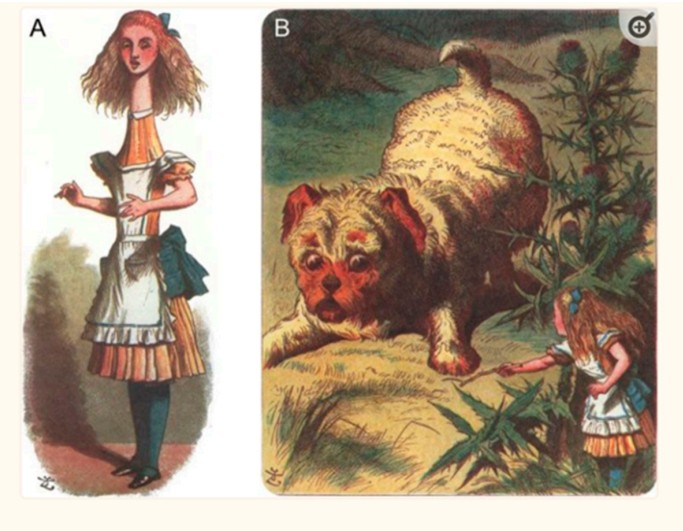

**Figure 2.** (**A**) Alice experieces partial macrosomatognosia, and (**B**) Alice experiences total-body microsomatognosia. Illustrations by John Tenniel (1890). Note: Alice in Alice in Wonderland. Adapted from "Alice in Wonderland Syndrome," by Blom, J. 2016, *The Journal of NIH Research*, V6(3): 259–270. Copyright 2016 Adapted with permission.

## 2. Case

A neurodevelopmentally normal patient arrived at the physician's office with a chief complaint of pulsating headaches on the left side of his head. The pain lasted over 24 h and was reoccurring for about 2 months. Before the headaches, he experienced some nausea, photophobia, and phonophobia. He described things around him as being of varying sizes and shapes. He expressed how the fingers on his right hand appeared smaller than those on his left hand. The visual disturbances had been occurring for over one year and they were becoming more prevalent in the last few months. The distortions lasted 45 min. His past medical history was unremarkable other than his mother and sister suffering from migraine headaches. His physical exam was ordinary, and his psychiatric exam yielded no findings. According to the psychiatric examination, he was conscious and well-oriented to time, place, person, and situation. Neurological exams were normal, and no pathologies were noted on brain MRI or EEG. A complete blood test panel was performed, and all results were within normal limits. The Structured Clinical Interview for DSM-IV Axis 1 Disorder was administered. Beck Anxiety Inventory score was 13 and the Hamilton Depression Rating Scale (HDRS) score was 17. He was treated with 500 mg/day of valproic acid, and his complaints improved. During follow-up, his daily dose of valproic acid was increased to 1000 mg/day, and the complaints were completely eliminated [4].

Migraines with aura are primary headaches in which focal neurological complaints usually evolve over a period of 10 min. These headaches more often manifest with recurrent attacks. They can have visual symptoms, sensory symptoms, motor weakness, and/or speech disorders. These could range from flashing lights or spots to numbness and pricking.

Our subject in this case study had symptoms consistent with those of AIWS. In our subject, there was no history of psychiatric disorder, hallucinogenic drug use, or central nervous system infections. As AIWS can be seen in temporo–occipital or temporo–partial–occipital lesions, our subject did not demonstrate any of these lesions in MRI findings.

## 3. Discussion

Since 1955, about 170 cases of AIWS have been reported in the literature, with most subjects being less than 18 years old. Only a part of them fit Todd's description. It was found in a study of 3224 subjects between 13–18 years old, that the occurrence of micropsia and macropsia was 6.5% and 7.3% in males and females, respectively. This suggests that visual illusions in AIWS are not as infrequent as usually believed [5].

Migraine is one of the most common headache disorders affecting approximately 12% of the general population. Auras are the sensory symptoms (neurologic, gastrointestinal, and autonomic) that can occur before an episode. These symptoms can include flashes of light, blind spots, or tingling in the hands or face [6]. About 10% of all migraine sufferers do not experience auras. Migraine shares common genetic variant risks with depression. Specific clinical features of common migraine seem to be determined by genetic factors. A stronger family history of migraine is associated with lower age-at-onset, higher frequency, the number of medication days, and the migraine with aura subtype [7]. Most attacks are trailed by either hours or a full day of feeling unwell called a postdrome. The symptomatology of AIWS differs from that of schizophrenia spectrum disorder and other disorders exhibiting hallucinations. To understand what makes AIWS stand out from other perceptual syndromes and disorders, we need to realize that distortions differ from hallucinations and illusions in that they are neither newly formed percepts of something that is not there (hallucination), nor actual objects or scenes mistakenly judged to be something else (illusion) [8].

A study on migraines found that verapamil was reported to be 55% effective as a treatment, while valproic acid was determined to be 18% effective. Evidence suggests that valproic acid is favorable in the case of migraine-induced AIWS [9]. The three-month follow-up showed no signs of re-occurrence in the patient. Studies have shown that valproic acid is a less effective treatment regimen for migraines, however in the case of this patient, valproic acid showed promising results. Definitive diagnostic exams aid practitioners in

keeping AIWS from being underdiagnosed. The fleeting nature of the episodes evidently makes it difficult for doctors to study this syndrome, but future research should help better understand its effects. The diagnosis of AIWS is made by complete history review and physical examination (involving neurologic, otologic, and ophthalmic tests) [10]. There is evidence of reduced blood circulating throughout the brain in patients who exhibit this syndrome on CT brain scans. Compared to a normal subject, patients displaying micropsia during an MRI will display a decrease in activation of the cortical regions in the brain. Migraine aura is a transient neurological symptom that most commonly involves the visual fields and occurs before the headache phase [11].

The genetic basis of migraine is complex, and there are some discrepancies as to which loci and genes are involved in the pathogenesis. Theories suggest that they are based on several genetic sources at varying genomic locations working in tandem with environmental factors. Understanding the specific genes in an individual with migraines would determine the targeted prophylactic treatment. Studies show that particular mutations in transmembrane genes in the brain are indicative of migraine headaches. The mutated genes stimulate voltage-gated calcium channel regulation.

Even though the pathogenesis of migraines is not fully understood, it is evident that there is a combination of both peripheral and central nervous system involvement. The AIWS may also be due to the abnormal blood stream in the parts of the brain that process visual perception and texture. This condition may either be transient or permanent with multiple aetiologies [12]. According to research, headaches are produced by vasodilation and aura by vasoconstriction. Around the trigeminal nerve fibers around the vessels of the pia mater, there is the activation of trigeminal afferents by neuronal pannexin-1 mega channel opening followed by activation of caspase-1, and the release of proinflammatory mediators coupled with the activation of NF-kB (nuclear factor kappa-B) [13]. This pathway can describe cortical depression and the activation of trigeminal nociception. AIWS is a diagnosis of exclusion. From a pathophysiological viewpoint, lesions (structural or functional) in different parts of the perceptual network can cause perceptual distortions, e.g., area V4 (hyperchromatopsia) and area V5 (akinetopsia) [14].

## 4. Prognosis

It is imperative that physicians and practitioners are educated on the relevance of this condition. A diagnosis can easily be misinterpreted due to its similarity to other psychiatric and neurologic conditions. The pathophysiology of AIWS is a definite illustration of the range of manifestations that transcends from pathologies of the nervous system and the surreal sensory-perceptive experiences of people afflicted by this syndrome. Various neurons have been demonstrated to play a role AIWS.

While no definitive examination is diagnostic of AIWS, the range of potential symptoms is expansive. When diagnosing AIWS, practitioners should review the symptoms the person is experiencing. The list of possible symptoms to include illusory changes in the size, distance, or position of stationary objects in the subject's visual field; illusory feelings of levitation; and illusory alterations in the sense of the passage of time [15]. Testing should include neurological and psychiatric consultations to assess mental status, routine blood testing, and brain scans to test electrical activity.

## 5. Conclusions and Future Recommendations

The unknowns of AIWS are still puzzling to many practitioners and scholars around the world. There is no definitive medication regimen for AIWS. Based on the individual's particular case, physicians may prescribe migraine preventive medication, antibiotics, or antivirals. Within the last five years, AIWS has begun to receive scientific attention once more. This is vastly due to the heightened interest in exploring the brain network and neuroimaging diagnostic advances in the field of neuroscience, psychiatry, and neurology. The hope for the future is that more light is shed on this syndrome, and it gets a proper classification in the Diagnostic and Statistical Manual of Mental Disorders (DSM-5) [16].

**Funding:** This research received no external funding.

**Institutional Review Board Statement:** Not applicable.

**Informed Consent Statement:** Obtained by subject in study.

**Data Availability Statement:** Not applicable.

**Conflicts of Interest:** The author declares no conflict of interest.

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
