# Peer review of "The Perplexing Mental Health Comorbidity of Alice in Wonderland Syndrome (AIWS): A Case Study"

_2673-5318, doi:10.3390/psychiatryint4010005_

Round 1

Reviewer 1 Report

The main question is about the rare case of the Alice in Wonderland Syndrome. The author recorded the case in details and explained the features of this illness deeply. Therefore, the author answered the main question completely. 
It is relevant as a part of the mental health during migraine attack, it is interesting as many doctors and professional health have no idea about this kind of issue.  

I assessed the information in the paper especially in the introduction and discussion, and I found it is original.    It adds new and updated information about rare case, to help doctors and other mental health professionals for diagnosis and potential management if they find similar case in future, it is well written with very minor issues.

The conclusion is relevant to the introduction and discussion and observational findings in this paper. 

Author Response

I agree with every point of the reviewer's response. The reviewer agreed that I explained the main cause of the paper in detail and answered the main question completely. I agree that they felt this is a compelling topic as it relates to migraines and most doctors are not aware of this syndrome so it brings new knowledge to the field.  Also, I agree with the reviewer that the paper is original and relevant.   In conclusion, the reviewer had only positive things to say and there were no areas for correction for this reviewer.

Reviewer 2 Report

Dear author,

   It is noteworthy that there is an article about a rare syndrome. In the summary section, I suggest shortening the historical information, restructuring the case report (inserting information about the summary and differential diagnosis), and emphasizing the final sentences. The case report needs to be written in more detail. It is appropriate to detail the patient's past information, to write down the neurological and psychiatric examination and imaging findings in more detail, according to which criteria the migraine diagnosis is made, and to expand the treatment part. "she was conscious, and well oriented to time, place, person, and situation." sentence she should be corrected. All text should be carefully reviewed regarding abbreviations.

Author Response

Hello,

I went ahead and made some of the changes as per the reviewer's comments.  More details were added to the case report.  Additionally, I went ahead and gave more detail about the subjects background, his neurological examination and psychiatric examination and included more detail about types of headaches.  Also describe what what meant by "he was conscious and well oriented to time, place, person, and situation"

Reviewer 3 Report

ן suggest adding  more  references 

Author Response

I agree with what the author suggested which was to add more references.  I went ahead and added a few more references.

Reviewer 4 Report

This is a case report of an adult case with Alice-In-Wonderland Syndrome, a rare neuropsychiatric condition, characterized by occasional episodes of bizarre visual illusions and spatial distortions. 

This syndrome is rare, therefore, it is worth to be shown in detail. However, the title is exaggerated. It is unclear what is novel and informative in this case report. Moreover, the manuscript is written like a historical review on this syndrome. 

Therefore, I have to comment that this manuscript doesn't contain any reportable information. I hope my comments will be a clue to refine the author's standpoint on this report.

Author Response

I went ahead and made corrections as per reviewers feedback.  I have selected a title that appeals more to the work presented as the reviewer thought the title was exaggerated and unclear.  I believe the new topic is more conducive to the work presented.  I added more in order to explain the history of headaches and more reportable information as indicated.

Round 2

Reviewer 4 Report

Editors' final decision will depend on the other excellent reviewers' comments and the authors' responses.

Thank you for this opportunity.